# Small molecule inhibitors of RAS-effector protein interactions derived using an intracellular antibody fragment

Camilo E. Quevedo[1], Abimael Cruz-Migoni [1,2], Nicolas Bery[1], Ami Miller[1], Tomoyuki Tanaka[3,8], Donna Petch[3], Carole J.R. Bataille[4], Lydia Y.W. Lee[5], Phillip S. Fallon[5], Hanna Tulmin[1,9], Matthias T. Ehebauer[1,10], Narcis Fernandez-Fuentes [2,6], Angela J. Russell[4], Stephen B. Carr[2,7], Simon E.V. Phillips [2,7] & Terence H. Rabbitts[1]

Targeting specific protein–protein interactions (PPIs) is an attractive concept for drug development, but hard to implement since intracellular antibodies do not penetrate cells and most small-molecule drugs are considered unsuitable for PPI inhibition. A potential solution to these problems is to select intracellular antibody fragments to block PPIs, use these antibody fragments for target validation in disease models and finally derive small molecules overlapping the antibody-binding site. Here, we explore this strategy using an anti-mutant RAS antibody fragment as a competitor in a small-molecule library screen for identifying RAS-binding compounds. The initial hits are optimized by structure-based design, resulting in potent RAS-binding compounds that interact with RAS inside the cells, prevent RAS-effector interactions and inhibit endogenous RAS-dependent signalling. Our results may aid RAS-dependent cancer drug development and demonstrate a general concept for developing small compounds to replace intracellular antibody fragments, enabling rational drug development to target validated PPIs.

---

[1] Weatherall Institute of Molecular Medicine, MRC Molecular Haematology Unit, University of Oxford, John Radcliffe Hospital, Oxford OX3 9DS, UK. [2] Research Complex at Harwell, Rutherford Appleton Laboratory, Didcot, Oxon OX11 0FA, UK. [3] Leeds Institute for Molecular Medicine, St James University Hospital, Leeds LS9 7TF, UK. [4] Chemistry Research Laboratory, 12 Mansfield Rd, Oxford OX1 3TA, UK. [5] Domainex, Chesterford Research Park, Little Chesterford, Saffron Walden CB10 1XL, UK. [6] Institute of Biological, Environmental and Rural Sciences, University of Aberystwyth, Aberystwyth SY23 3EB, UK. [7] Department of Biochemistry, University of Oxford, South Parks Road, Oxford OX1 3QU, UK. [8] Present address: Research & Development, Sanofi K.K., Tokyo Opera City Tower, 3-20-2, Nishi Shinjuku, Shinjuku-ku, Tokyo 163-1488, Japan. [9] Present address: Wellcome Trust Centre for Human Genetics, Roosevelt Drive, Oxford OX3 7BN, UK. [10] Present address: Oxford Drug Discovery Institute, Nuffield Department of Medicine Research Building, University of Oxford, Old Road Campus, Oxford OX3 7FZ, UK. These authors contributed equally: Camilo E. Quevedo, Abimael Cruz-Migoni. Correspondence and requests for materials should be addressed to T.H.R. (email: terence.rabbitts@imm.ox.ac.uk)

There are at least two problem areas in devising therapeutics to intracellular targets in disease. Most are not enzymes per se for which active site inhibitors can be derived, but rather their function is mediated by specific protein–protein interactions (PPIs)[1]. This has led to the development of macromolecules like intracellular antibody fragments[2–4] (herein referred to as macrodrugs, distinct from conventional drugs)[5] that fold and interact with targets in the intracellular environment and can blockade PPI due to higher relative affinity scores compared with natural PPI partner[6]. Thus intracellular antibodies or peptide aptamers[7] can easily be selected with high affinity and be used for target validation by interrogating relevant preclinical models for effects on the specific disease, such as a mutant RAS in cancer[3]. However, devising methods to internalize these macrodrugs into cells to achieve the function has been elusive. Small-molecule drugs have opposite innate properties to macrodrugs. They can readily penetrate cells, but they are thought to lack the ability to interfere with PPIs because of low affinity and low surface area interaction[8–10], although examples of compounds with effects against PPI have been described in recent years[11,12]. One way to bring together these various properties is to use macrodrugs that have been used for target validation to select small compounds that bind to the target at the same location and which would thus have the potential for hit to lead drug development (macrodrugs include a variety of macromolecules, ranging for instance from oligonucleotides, to mRNA to proteins). Human intracellular single-domain antibody fragments have been well characterised since the first example[13]. The binding site of a variable region domain comprises about 600 Å$^2$ [14] and is the minimal region of an antibody-binding site recognizing an antigen[15]. This is a very small region equivalent to less than 500 daltons[16] and can narrow down the protein target area in competition screenings. Searching in smaller areas will increase the chances of detecting small molecules (within the Lipinski rules[17]) with similar properties as the previously validated antibody fragments.

The RAS family of proteins is among the most frequently mutated in human cancers[18,19], with *KRAS* mutations found in almost all pancreatic tumours, about 40% of colorectal tumours and about 30% of lung adenocarcinomas[20] http://cancer.sanger.ac.uk/cosmic. Reagents that block these RAS-effector interactions have thus far largely been macromolecules ranging from cyclic peptides[21] to antibodies[22,23] or antibody fragments[3,4] and, from a number of approaches targeting the RAS family of proteins with small compounds[24–32], only two have shown direct RAS-effector interface inhibition[31,32]. We have characterized an antibody fragment, using intracellular antibody capture technology[33,34], that specifically binds to the activated forms of HRAS, KRAS and NRAS with optimal binding properties (low Kd, high $K_{on}$ and low $K_{off}$) and inhibits tumour growth in xenograft models[35] even when mutations in other proteins are present in the cells[3]. Crystallisation with the antibody fragment yielded an scFv-HRAS$^{G12V}$ crystal complex, where the binding site of the antibody fragment was identified and its RAS-effector inhibitory properties were understood[36].

Here, we employ the intracellular antibody to identify a compound from a chemical fragment library that binds to RAS proteins adjacent to the effector binding region. High-resolution crystal structures show the binding of this compound to mutant RAS, and this structural information facilitates hit to lead development. Our structure-based approach has produced compounds that bind to mutant KRAS with sub-μM affinity that interfere with RAS-effector PPI and endogenous RAS-dependent signalling in human cancer cells, and show a correlation between in vitro binding and potency in the cells. Our work demonstrates that intracellular antibody fragments can be used as progenitors in drug development programmes to arrive at high-affinity compounds binding to mutant KRAS. This approach can be generalised to any PPI in a disease.

## Results

**SPR with anti-RAS antibody identifies RAS-binding compounds.** The initial SPR screening strategy is outlined in Fig. 1a–c, in which compounds from a fragment-based, small-molecule library were counter-screened for the binding to GTP-bound (Fig. 1a) or GDP-bound (Fig. 1b) HRAS. The chemical fragment library, comprising 656 compounds, was screened using a BIAcore T100 with single-point analysis of each compound at 200 μM (Fig. 1d). Putative RAS-binding compounds were individually re-tested using GST-HRAS$^{G12V}$-GTPγS and wild-type GST-HRAS-GDP. One compound, out of 26 initial hits, with a quinoline core (Abd-1) showed preferential binding to HRAS$^{G12V}$-GTPγS (Fig. 1e, g). The dose response of Abd-1 (3 to 2000 μM range) was tested in cSPR using immobilized GST-HRAS$^{G12V}$-GTPγS, GST-HRAS$^{G12V}$-GDP or GST-HRAS$^{G12V}$-GTPγS-anti-RAS single chain Fv (scFv)[3] (Fig. 1e–g). Abd-1 bound to GST-HRAS$^{G12V}$-GTPγS with a dose response (Fig. 1e). In contrast, no binding was observed when HRAS$^{G12V}$ was in complex with the anti-RAS scFv (Fig. 1f) or with the GDP-bound form of HRAS (Fig. 1g). The Kd for Abd-1 was estimated greater than 370 μM against HRAS$^{G12V}$ GTPγS (Supplementary Fig. 1).

The chemical structure of Abd-1 is shown in Fig. 1 (MW:238.25, clogP 2.33, solubility 140 μM). We sought analogues with improved biophysical properties and potentially better Kd. One candidate analogue (Abd-2, Fig. 1h) was identified (Mw 259.26; logP 1.83) with a solubility of 446 μM. Abd-2 has a benzodioxane bicyclic core with furanyl amide as the right-hand side of the molecule. SPR analysis showed that compound Abd-2 binds to both HRAS$^{G12V}$-GppNHp and KRAS$^{G12V}$-GppNHp (Fig. 1h shows KRAS binding). Abd-2 is a racemic mixture and the enantiomers (compounds Abd-2a and Abd-2b) were tested for binding to KRAS$^{G12V}$-GppNHp by SPR (Fig. 1i, j). Both enantiomers bind to KRAS$^{G12V}$ with similar properties. A truncated analogue of Abd-2, designated as Abd-3, also binds to KRAS$^{G12V}$-GppNHp, but with lower affinity than Abd-2 (Fig. 1k), showing the intrinsic binding properties of the benzodioxane core.

We measured the dissociation constant of Abd-2 using waterLOGSY[37,38] showing a Kd of 235 μM (Supplementary Fig. 2). We also confirmed that Abd-2 reproduced preferential binding to the activated form of KRAS$^{G12V}$-GppNHp compared with the form KRAS-GDP (Supplementary Fig. 3a) and no longer binds to KRAS$^{G12V}$-GppNHp in the presence of anti-RAS scFv (Supplementary Fig. 3b).

**Antibody-derived compounds bind to KRAS near the switch regions.** Protein crystallography was used to determine the KRAS-binding site for Abd-2 and for the truncated analogue Abd-3. Two different mutant KRAS-GppNHp proteins, G12D and Q61H were crystallised, and the crystals were soaked with compounds to yield high-resolution structures (Fig. 2, Supplementary Table 1). Compounds Abd-2 and Abd-3 bind in a pocket adjacent to the RAS switch regions I and II (Fig. 2a, b) that has been previously described as a site of other small-molecule binding[24,25,39] Residues lining the binding pocket in KRAS$^{Q61H}$ include K5, L6, V7, S39, D54, I55, L56, Y71 and T74 (Supplementary Fig. 4). Electron density for Abd-2 was identified only with KRAS$^{Q61H}$-GppNHp (crystal form I) and in five of the six KRAS molecules in the asymmetric unit (the ligand density in chain B is used for illustration in Fig. 2a). The electron density is stronger around the benzodioxane core, with the furanyl amide substituent less well defined (Fig. 2a). No H-bond interactions

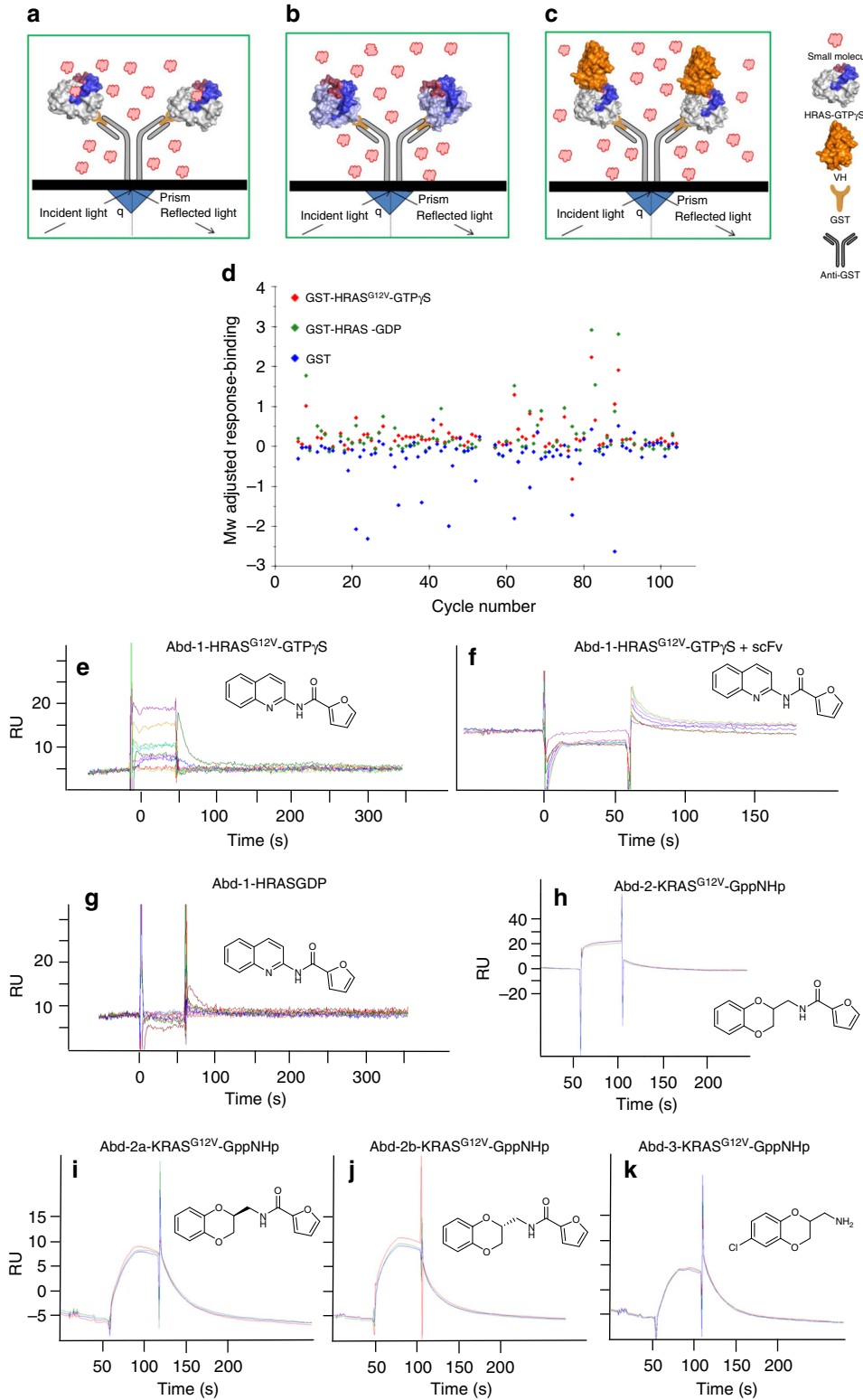

were observed between the compound and protein, with only van der Waals contacts with K5, L6, V7, E37, D38, S39, D54, I55, L56, G70, Y71 and T74. The electron density around furanyl amide provides an orientation marker for the binding pose of this compound and also the explanation of the competitive effect of anti-RAS antibody fragment on its binding (see below).

Analysis of the compound Abd-3–KRAS^Q61H complex (crystal form I) shows the best electron density in chain B (Fig. 2b and Supplementary Fig. 4). The ligand terminal $NH_3^+$ makes H-bond

interactions (salt bridge) to D54 carboxylate and to a well-defined water molecule (430) that is in turn also co-ordinated by an H-bond to D54 carboxylate and the hydroxyl group of S39. In contrast, Abd-3 has a rather different binding geometry in the KRAS^G12D crystal structure (crystal form II, Fig. 2c), as it is stabilized by two H-bonds via the ligand terminal $NH_3^+$, one with E37 carboxylate and the other with the main chain carbonyl oxygen of D119 from a neighbouring protein in the KRAS^G12D crystal lattice. Thus, both interactions stabilize the terminal

**Fig. 1** Competition SPR identifies RAS-binding compounds. **a, b** The SPR screen involves a differential binding approach to identify compounds binding to activated, GTP-bound mutant RAS (HRAS$^{G12V}$), but not GDP-bound HRAS, indicated in (**a**) and (**b**), respectively. **c** Schematic representation of the cSPR approach. Anti-GST polyclonal goat antibody was captured on a CM5 SPR chip, and GST-RAS proteins were captured with anti-GST. Compounds that bind to a target protein (in this case HRAS) can be challenged with binding to the target protected by the high-affinity antibody fragment. If the binding regions coincide, the compound will not bind to the target. **d** A chemical fragment library of 656 compounds was initially screened as single points at 200 mM using in the four channels (Fc) of a Biacore T100: Fc1: reference cell; Fc2: red diamond GST-HRAS$^{G12V}$-GTPγS (active form of HRAS target); Fc3: green diamond GST-HRAS wild-type protein-GDP (inactive form of HRAS); Fc4: blue diamond recombinant GST only. **e–g** The RAS-binding compound Abd-1 was shown by SPR (dose-response sensogrammes using 3.9, 7.81, 15.6, 31.3, 62.5, 125, 250, 500, 1 and 2 mM compounds) to bind to mutant HRAS$^{G12V}$-GTPγS (**e**), but not to HRAS$^{G12V}$-GTPγS-anti-RAS scFv complex (**f**) or HRAS-GDP (**g**). **h–k** Analogues of the initial hit were identified and shown to bind to KRAS$^{G12V}$-GppNHp at 100 μM, run in triplicate

---

$-CH_2NH_3^+$ functionality in a different orientation from the one observed for KRAS$^{Q61H}$, but it is directly influenced by the packing in crystal form II (Fig. 2b, c and Supplementary Table 1).

The mechanism by which the antibody fragment inhibits the binding of compounds was revealed by comparing the structures of HRAS$^{G12V}$-Fv[3] with KRAS$^{G12V}$-Abd-2 (Fig. 2d). Superimposing the complexes shows that the CDR2 region of the VH is oriented towards the binding pocket, where benzodioxane interacts with the side chain of K56 in CDR2, partly overlapping with the furanyl substituent (Fig. 2d). Structure analysis shows that intracellular antibody does not cause structural changes in HRAS$^{G12V(3)}$ and the compound-binding pocket remains accessible (Supplementary Fig. 5). Therefore the competition occurs by displacement rather than allosteric changes in RAS protein.

**A chemical series from the antibody-derived compound Abd-2.** The initial hit antibody-derived compound gave us the anchor binding site on KRAS, from which to develop compounds to compete with RAS-effector PPIs. A chemical series was developed and representative structures are shown in Fig. 3a (Abd-4, 5, 6 and 7). High-resolution crystallography showed that all compounds bind in the same hydrophobic pocket near switch I (Fig. 3b, c and Supplementary Table 2) and with increasing potency (Supplementary Fig. 6). In the development of the series, efforts were focussed on the right-hand side (RHS) of the molecule (i.e. substituents of the dioxane), since crystallography consistently showed electron density predominantly around the benzodioxane core. Compound Abd-4 was extended from the amide group, but KRAS$^{Q61H}$ crystal soaking only showed electron density in two of the six chains (the ligand density in chain B is illustrated in Fig. 3, and only around the benzodioxane core (Fig. 3b, c). No H-bond interactions were observed between the benzodioxane rings and protein, with only van der Waals contacts with K5, L6, V7, E37, D54, I55, L56, G70, Y71 and T74. Therefore, efforts were directed to substitutions on the left-hand side (LHS), on the benzene ring of the benzodioxane main core, targeting interactions with K5, D54 or S39.

Aromatic substituents at position 8 in Abd-5 increased the level of electron density observed around the compound, and it was identified in all six chains of the asymmetric unit (chain F is illustrated in Fig. 3). Crystal soaking of KRAS$^{Q61H}$ with Abd-5 showed the new pyridine ring leading to the side of the switch I region of KRAS (Fig. 3c). This compound forces D54 to rotate to allow the third ring to be accommodated in a pocket formed by S39, Y40, D54, I55 and L56. No H-bond interactions were identified between Abd-5 and the protein, and the only interactions identified were van der Waals contacts to K5, L6, V7, S39, Y40, R41, D54, I55, L56, G70, Y71 and T74. No interactions were observed from the RHS of the molecule.

Analysis of crystals soaked with Abd-5 indicated that exploration around the pyridine ring in positions 3 to 5 could gain extra interactions in the KRAS switch I region. Abd-6 included a third

ring (benzene) at position 3 of the pyridine ring in Abd-5 and a tetrahydropyran-4-carboxamide substituent in place of the benzamide derivative on the right-hand side (Fig. 3b, c). Soaked crystals showed electron density for the new aniline-based functionality, but no H-bond with the protein in any of the four chains where the compound was identified. The only interactions observed were van der Waals contacts to K5, L6, V7, S39, Y40, R41, D54, I55, L56, G70, Y71, T74 and G75, and also with amino acid residues from a neighbouring protein within the unit cell. The binding affinities of Abd-5 and Abd-6, measured using NMR-binding detection method Carr–Purcell–Meiboom–Gill (CPMG), a relaxation dispersion technique (RD) using proton NMR's of the ligand molecule as sensors for binding and unbinding kinetics and without any isotope labelling. In CPMG, signal reduction occurs with increased protein concentration[40,41]. This showed that Abd-5 and Abd-6 had improved affinities of 220 nM and 38 nM (respectively, Supplementary Fig. 6b and d)

An important feature in Abd-5 and Abd-6 was the lack of interactions between KRAS$^{Q61H}$ and the benzamide derivative (Abd-5) or the tetrahydropyran-4-carboxamide (Abd-6), substituted from the original furan group in Abd-2. The RHS of Abd-6 was removed to reduce the molecular weight, but to maintain or increase the interaction with the protein. The new analogue, Abd-7, gave improved crystallography results with full electron density for the compound (Fig. 3a–c) and with van der Waals contacts to K5, L6, V7, S39, Y40, R41, D54, I55, L56, G70, Y71, T74 and G75. Abd-7 was found in five out of six protein chains. Interactions to residues from a neighbouring protein within the unit cell were also identified, same as Abd-6.

The protein-binding capabilities of the most potent of this series of compounds (Abd-5, Abd-6 and Abd-7) was confirmed in an orthogonal assay (waterLOGSY) using KRAS$^{G12V}$-GppNHp (Supplementary Fig. 6). The spectra for each compound show that all the proton peaks have reverse polarity in the presence of the KRAS protein (Supplementary Fig. 6a, c and e). The values of 220, 38 and 51 nM were found, respectively, for compounds Abd-5, Abd-6 and Abd-7 (Supplementary Fig. 6b, d, f). The ligand efficiencies values (each approximated to 0.3), molecular weights and solubilities for these three compounds are shown in Supplementary Fig. 6g.

**KRAS-binding compounds inhibit RAS PPIs in cell assays.** We have investigated the ability of the chemical series to inhibit RAS-effector PPI using cell-based assays involving a RAS-effector BRET2 assay, the effect on downstream biomarker phosphorylation and tumour cell viability. A RAS PPI BRET2 toolbox has been developed to evaluate the in vivo effect of compounds on RAS-effector PPI of compounds[42]. This assay comprises transfection of HEK293T cells with plasmids to express BRET donor (fusions of either K, N or HRAS, including a carboxy-terminal farnesylation signal tetrapeptide, with *Renilla* luciferase variant 8, Rluc8) and acceptor molecules (fusions of effector proteins with GFP[2]), and permits the assessment of inhibitors of RAS-effector

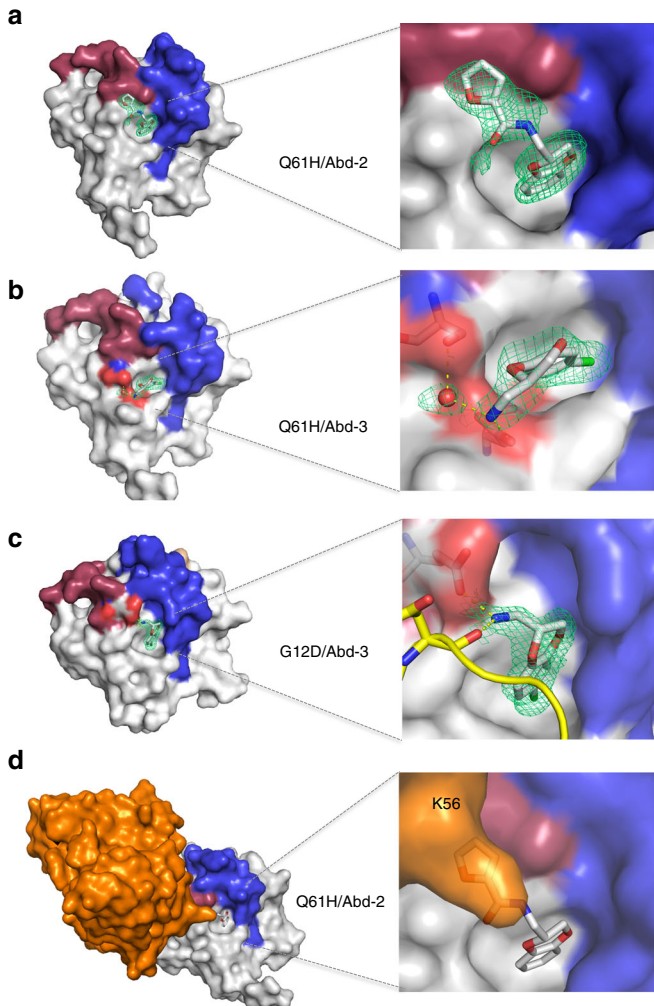

**Fig. 2** Crystal analysis shows how compounds bind and VH competition results. KRAS$^{Q61H}$ or KRAS$^{G12D}$ protein crystals were soaked with compounds, and X-ray diffraction data were collected for determining the binding modes of the compounds. **a** KRAS$^{Q61H}$-GppNHp soaked with Abd-2. The expanded view of the binding region of this compound (right hand panel) shows clear electron density (*2mFo-DFc* maps contoured at 1.0 r.m.s. green) attributed to the benzodioxane and furanyl amide parts of the compound. **b, c** Crystal structures and electron densities for Abd-3 soaked into KRAS$^{Q61H}$-GppNHp or KRAS$^{G12D}$-GppNHp, respectively. The chlorine atom in Abd-3 is depicted in green. The Abd-3–KRAS interactions differ in two mutants, but the H-bond to a neighbouring molecule in the crystal lattice for G12D means that the Q61H complex is unencumbered by the crystal contacts (**b**). The switch I/II regions are coloured in red and blue, respectively, are defined here as 30–38 (switch I) and 60–76 (switch II). **d** Explanation for the competition of compound Abd-2 binding to RAS by steric hindrance. The left-hand panel shows a surface representation of mutant HRAS$^{G12V}$-GppNHp (light blue) and the anti-RAS VH from the Fv depicted in orange. The left-hand panel is the surface representation is the KRAS$^{Q61H}$-GppNHp structure soaked with Abd-2, with anti-RAS VH superimposed on KRAS$^{Q61H}$-GppNHp. The expanded right-hand representation shows the predicted steric hindrance between VH and the compound, in particular VH CDR2 residue K56 (transparent, orange representation). Although the K56 side chain is flexible, it is prevented from rotating away from the clash with Abd-2 by steric hindrance with neighbouring regions of KRAS

interaction. We determined the effect of Abd-7 on the interaction of KRAS$^{G12D}$ and with PI3Kα and PI3Kγ, CRAF RAS-binding domain (RBD) and RALGDS RAS-associating domain (RA) compared with the low-affinity Abd-2 Fig. 4a). Abd-2 has no effect on the BRET signal over a range of 5–20 μM, while Abd-7 reduces the BRET signal at 5, 10 and 20 μM for all of the RAS-effector PPIs tested. A similar inhibitory effect of Abd-7, but not Abd-2, was observed using five different glycine 12 mutations of KRAS interacting with the full-length CRAF-GFP[2] fusion (Fig. 4b). Finally, we tested the efficacy of Abd-7, compared to Abd-2, in the BRET assays using other RAS family members, either NRAS$^{Q61H}$ or HRAS$^{G12V}$, interacting with PI3Kα and PI3Kγ, CRAF-RBD or full-length CRAF and RALGDS-RBD (respectively, Fig. 4c, d). Abd-7 interferes with all mutant RAS family member PPIs in this transfection assay. However, in this type of assay, protein expression levels are difficult to control due to the transient transfections, and the percentage of signal reduction may vary between repetitions or between different RAS-effector partners. The generation of stable cell lines could reduce the differences on protein expression.

In order to assess the effect of our compounds on endogenous RAS signalling, we analysed phosphorylation of AKT (downstream of RAS–PI3K signalling) and phosphorylation of ERK (downstream of RAS–RAF signalling) using DLD-1 cells (a colorectal line with KRAS$^{G13D}$ mutation) and H358 cells (a NSCLC with KRAS$^{G12C}$ mutation). The cells were starved for 24 h, followed by incubation for 3 h with an increasing dose of Abd-7 (range 2–20 μM) and stimulated EGF for 10 min to activate KRAS signalling. Protein levels were determined by Western analysis (Fig. 5a, b). While no change of endogenous AKT or ERK 1/2 was observed, inhibition of AKT phosphorylation (assessed with anti-pAKT Ser473 antibody) was observed in both cell lines, with response initiating between 2 and 5 μM, more than 50% at 10 μM and complete inhibition at 20 μM. No change was observed in cyclophilin B (as a loading control on the Western blots) (Fig. 5a, b). Similar dose responses were obtained for inhibition of ERK1/2 phosphorylation by Abd-7 (assessed with anti-pERK1/2 antibody) treatment of the cells. EGF stimulates pERK production without affecting ERK1/2 levels, and diminution of pERK levels was found with a similar dose range of Abd-7 (Fig. 5a, b). We confirmed that Abd-7 is not directly binding to kinase proteins themselves, using a binding assay with a panel of kinase proteins, including RAF, MEK and PI3K (Supplementary Table 3), endorsing the conclusion that control of phosphorylation depends on the direct binding of Abd-7 to RAS protein rather than to other protein kinases. These cell assays show that the lead compound Abd-7 is cell permeable and it exerts its inhibitory function by interfering with PPI of RAS and effectors. Independent evidence for cellular uptake was obtained using a Caco-2 cell permeability assay (Papp A-B/B-A) 7.03/8.46 (10$^{-6}$ cms$^{-1}$).

We carried out cell viability studies using a monolayer culture system with two human cancer cell lines with different RAS family member mutations (DLD-1 KRAS$^{G13D}$ and HT1080 NRAS$^{Q61K}$). A dose response was carried out with Abd-2, Abd-4, 5, 6 and 7. Figure 5c (DLD-1) and 5d (HT1080) show cell viability in the presence of 0–20 μM compound after 72 h (Supplementary Fig. 7 shows viability at 24 and 48 h). The lowest-affinity compounds, Abd-2 and Abd-4, show no effect on cell viability even at the highest doses (these compounds also do not show activity in the RAS biosensor BRET assay). Compounds Abd-5 and Abd-6 have an effect on the viability (with Abd-5 showing a greater effect on DLD-1 viability); but even at 20 μM

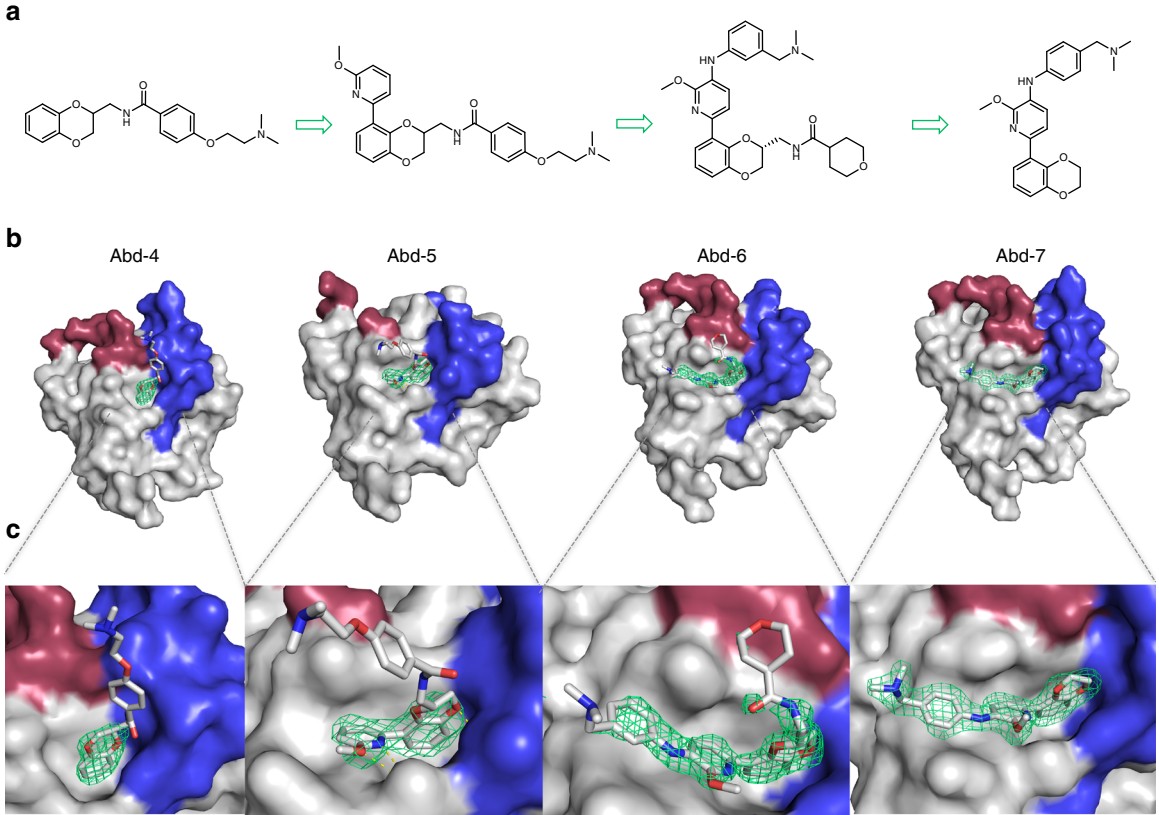

**Fig. 3** RAS-binding series development from antibody-derived initial hit. Representative examples of KRAS-binding compounds Abd-4 to Abd-7 guided by structural biology information. **a** Chemical structures of the chemical series with numbering on different rings. **b** Crystal structures with the mode of binding of each compound to KRAS<sup>Q61H</sup>-GppNHp (shown in grey) in the pocket close to the switch regions I (red) and II (blue). Each new analogue has extended its interaction with the protein, reaching to parts of the switch I region. **c** Expanded view to the compounds binding to KRAS, with the electron density identified in the crystallographic experiments depicted as a green mesh (*2mFo-DFc* maps contoured at 1.0 r.m.s)

and after 72 h, there are surviving cells following incubation with either compound. The most potent compound is Abd-7 (whose in vitro Kd is 51 nM) with an $IC_{50}$ of 8 μM in DLD-1 and 10 μM in HT1080 at 72 h (similar values for the $IC_{50}$ were found after 48 h (Supplementary Fig. 7f). These results showed that the ability of our chemical series to affect cancer cell viability relates to their binding properties, and that Abd-7 is the most cell-potent inhibitor affecting the viability of cancer cell lines in a single digit to low micromolar range.

## Discussion

Intracellular antibodies are highly specific research tools for target validation[3] nonetheless scFv, single domain intracellular antibodies (iDAbs)[13] and single camelid VH domains (nanobodies)[43,44] have not been implemented as therapeutic agents. An alternative use of intracellular antibodies is to select small molecules that bind to the same region of target antigens and can thence potentially be developed into drugs[5,45,46]. Our work describes the use of an intracellular antibody fragment that competes RAS-effector PPI[3], to select a compound in vitro from a chemical fragment library that binds to RAS family proteins at the cognate site. The method depends on a high-affinity interaction between the intracellular antibody fragments and to allow sustained interaction during the cSPR. This cannot be achieved using natural partner proteins (e.g. RAS-effectors) as these have low binding affinity with high $K_{off}$. We thus show in this work that a single domain intracellular antibody fragment, which was derived against mutant HRAS and used for target validation of RAS-effector PPIs, could be used in a competition assay to select RAS-binding compounds from a fragment library. This created a

chemical basis for development, by medicinal chemistry and SAR, of a compound series with improved properties, guided by high-resolution X-ray crystallography. We generated a soluble lead compound with low nM Kd, that is both cell permeable, was able to block RAS PPIs in human cancer cells, and to affect the viability of the cells. Although, various RAS inhibitors have been described in literature,[24–32,47,48] in this paper we describe an approach for the development of RAS-effector inhibitors by combining different elements i.e. (i) the hit identification via a method of intracellular antibody fragment competition; (ii) structure-based design via crystal structures (KRAS<sup>Q61H</sup>GPPNHP) and NMR (waterLOGSY and CPMG); (iii) identification of clear structure-activity relationship (SAR) for the development of potent compounds against KRAS and (iv) the proof of concept in mechanistic and phenotypic cell assays.

Our medicinal chemistry, guided by high-resolution crystallography allowed the improvement of our initial Abd-2, that binds in a previously observed pocket, through to the nM-binding compound Abd-7 (Fig. 3). X-ray crystallography using crystal soaking showed how the initial hits bind to KRAS and how the antibody fragment VH interferes with this binding (Fig. 2d, Fig. 6) Our guided approach has led to the identification of a novel series of compounds that interfere with the oncogenic function of RAS and whose binding features are confirmed. Different groups have described compounds binding to this same pocket, however, there has been no structure-based design of direct RAS-effector inhibitors done in that region. The difficulties of identifying a suitable crystal packing conditions of mutant KRAS to soak compounds contributes to the difficulties of generating SAR and, consequently, of generating highly potent

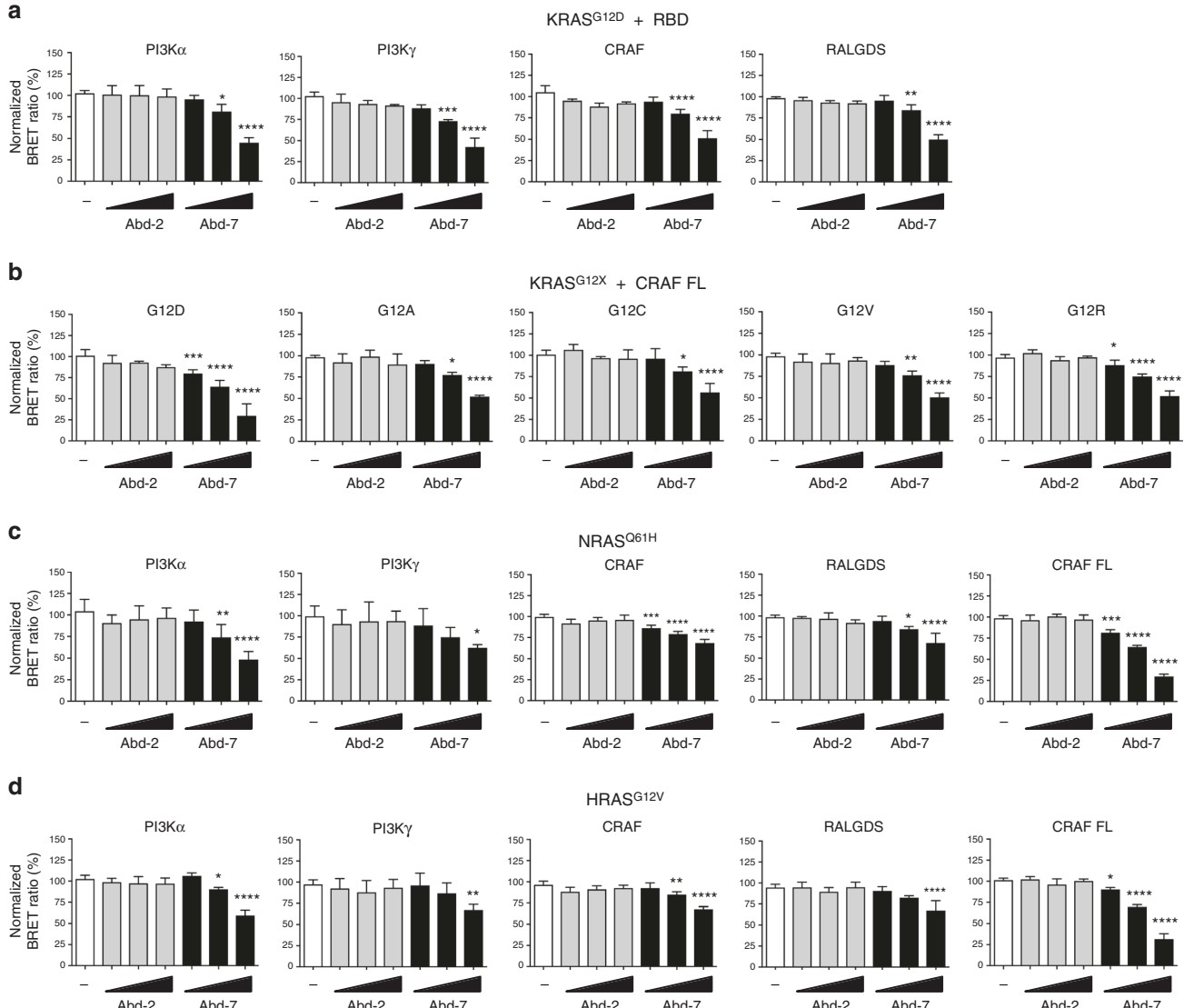

**Fig. 4** Abd-7 disrupting RAS-effector interactions. HEK293T cells were transfected with different BRET-based RAS biosensor expression vectors to evaluate the inhibition of RAS PPI in cells by compound Abd-7. Transfection vector encoded full-length RAS was fused to the donor molecule RLuc8 and the effectors fused to the acceptor molecule GFP[2]. **a** Effect of Abd-2 and Abd-7 on KRAS[G12D] interaction with PI3Kα, PI3Kγ, CRAF or RALGDS. The BRET signal is plotted as a % of control cells treated with DMSO only and dose response to 5, 10 and 20 μM of each compound. **b** Effect of Abd-2 and Abd-7 on the BRET signal from interaction of KRAS[G12] mutants (Rluc8-KRAS[G12]) and full-length CRAF (GFP[2]-CRAF FL). **c**, **d** Effect of Abd-2 and Abd-7 on the interaction of NRAS[Q61H] (**c**) and HRAS[G12V] (**d**) with various RAS effectors domain and with full-length CRAF. The BRET ratio corresponds to the light emitted by the GFP[2] acceptor constructs (515 nm ± 30) upon addition of Coelenterazine 400a divided by the light emitted by the RLuc8 donor constructs (410 nm ± 80). The normalized BRET ratio is the BRET ratio normalized to the DMSO negative and calculated as follows: (BRET_compound/BRET_DMSO) x 100, where BRET_compound corresponds to the BRET ratio for the compound-treated cells, BRET_DMSO to the DMSO-treated cells. Each experiment was repeated at least three times. Statistical analyses were performed using a one-way ANOVA followed by Dunnett's post-tests (*$P < 0.05$, **$P < 0.01$, ***$P < 0.001$, ****$P < 0.0001$). Where error bars are presented, they correspond to mean values ± SD of biological repeats (**a**–**c**)

compounds binding to pocket. We have solved this by exploiting and optimizing known crystallisation conditions for KRAS[Q61H] (PDB code 3GFT).

The initial fragment compound (Abd-2) was improved by introducing pyridine functionalities to position 8 of the benzene ring from the benzodioxane moiety, enhancing the Kd from μM to nM with increased ligand efficiency values from 0.26 of Abd-2 to 0.34 of Abd-7 (Supplementary Fig. 6). An expansion of the interactions is observed when Abd-4 was evolved to Abd-7 (Supplementary Fig. 8). One of our objectives was to show that intracellular antibody fragments could be used for chemical compound hit identification and to find compounds that emulate the inhibition of PPI shown by the antibody[3] in cells. We used a

BRET2-based assay to evaluate the intracellular inhibition of RAS PPI by Abd-7, compared to the low affinity compound Abd-2 and showed that Abd-7 impairs the PPI of various mutant KRAS proteins with PI3K, CRAF and RALGDS as well as NRAS[Q61H] and HRAS[G12V] but Abd-2 does not (Fig. 4). RAS-dependent signalling occurs via several kinase molecules, including phosphorylation of AKT (the RAS-PI3K pathway) and ERK (the RAS-RAF pathway). The Abd-7 compound inhibits phosphorylation of both downstream biomarkers (Fig. 4a, b) in colorectal cancer (DLD-1, KRAS[G13D]) and non-small cell lung cancer (H538, KRAS[G12C]) human cell lines in a dose dependent way, with reduction of AKT phosphorylation starting at 2 μM. These effects occur within 2 h of Abd-7 treatment while there is no effect on

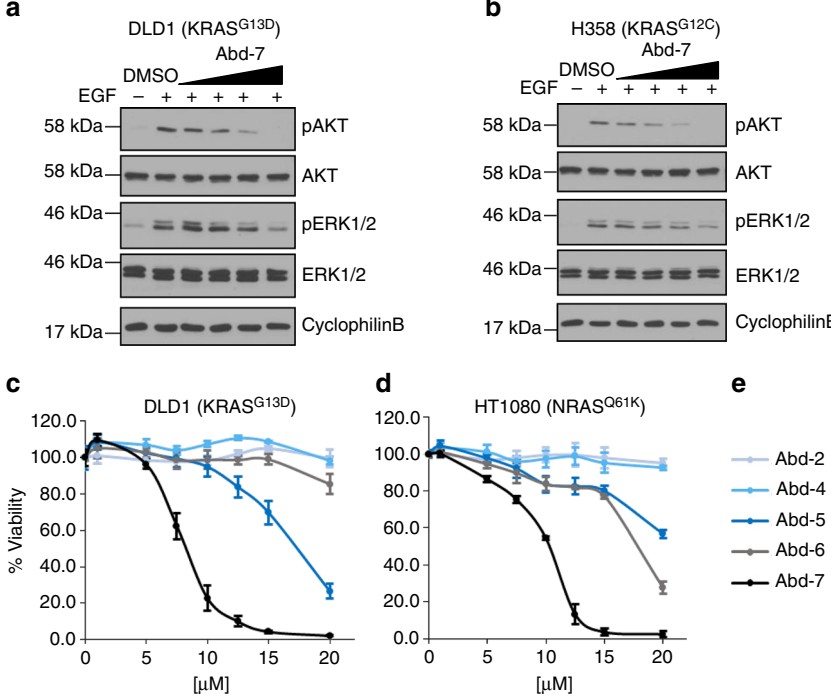

**Fig. 5** Abd-7 results on RAS-dependent signalling pathways and cell viability. **a, b** DLD-1 (**a**) or H358 (**b**) cells were serum-starved for 24 h, incubated in mono-layer with Abd-7 in a range from 2, 5, 10 and 20 μM for 3 h and stimulated with EGF 10 min. Proteins were extracted and separated by SDS-PAGE and transferred to membranes for Western analysis with anti-pAKT, anti-pERK, anti-pan AKT and anti-pan ERK. Anti-cyclophilin B antibody was the loading control. Signal was developed using standard ECL. **c–e** Effects of compounds Abd-2, Abd-4, Abd-5, Abd-6, and Abd-7 on the viability of human cancer cells lines was assessed for a 2-D culture of DLD-1 (mutant KRAS$^{G13D}$) cells and HT1080 cells (mutant NRAS$^{Q61K}$). The cells were treated with a dose range from 0 μM to 20 μ and incubated for 72 h when cell viability was assessed using CellTitreGlo. In each case, the data are normalized to cells treated with DMSO only. **c** is DLD-1 and **d** is HT1080. **e** shows the colour coding for the different compounds. Each experiment was repeated at least three times (**a**, **b**) and four times (**c**, **d**). Where error bars are presented, they correspond to mean values ± SD of biological repeats

the overall AKT or ERK protein levels. Abd-7 is, therefore, an inhibitor of RAS-effector PPI. The mechanism by which Abd-7 interferes with RAS-effector PPI is indicated from the crystal structure data. The binding of the CRAF, RALGDS and PI3K effectors to RAS is similar and very close to the binding site of Abd-7. In Fig. 6, RAS with bound Abd-7 is superimposed to structures of RAS-effector complexes showing that the terminal tertiary ammonium group, and adjacent phenyl ring, of Abd-7 overlaps with the bound effector structures. The competition effect of Abd-7 on the binding of the effectors with RAS in cells, can therefore be interpreted as steric hindrance of the lower affinity effectors by the higher affinity Abd-7. However, despite observing similar steric disruption of Abd-7 (Fig. 6), there are differences observed in the reduction of phosphorylation of pAKT and pERK in cells. These differences could be associated with the variations in affinity of the effectors towards KRAS. For example, Pi3K has an estimated affinity of 3.2 μM, while RAF has an affinity of 0.08 μM[6].

The susceptibility of cancer cells to the compounds in our chemical series was compared between the two human cell lines with different RAS family mutations (DLD-1 with mutant KRAS$^{G13D}$ and HT1080 with mutant NRAS$^{Q61K}$) using standard viability assays. The effect of the compounds on cell survival correlated with the potency of the compounds in binding using in vitro NMR and crystallography. The early compound (Abd-2), that was the direct descendent of the initial hit obtained using the cSPR, had not effect on either cell type, up to 72 h and the maximum concentration used (20 μM) (Fig. 5c, d). Compound Abd-7 displayed the highest IC$_{50}$ (approximately 8 μM in DLD-1 and 10 μM in HT1080 at 48 h, (Supplementary Fig. 7), had the lowest Kd (51 nM) and shows consistent inhibitory effects in the

cell-based BRET assay and the biomarker assay. The observed discrepancy between affinity (in vitro Kd) and efficacy (IC$_{50}$ in cells) is a known challenge that can be addressed through chemistry. A comparison between Abd-6 and Abd-7 affinity Kds and cell IC$_{50}$ demonstrates that by chemical modifications, one can increase cell potency while maintaining a similar-affinity Kd. Further increase of potency could be achieved by new analogue synthesis based on structural biology information as demonstrated in our work. In addition, it is also known that discrepancies between in vitro and in cellulo potencies can be associated with targets, such as RAS, with high conformational variability and allosteric modulators[49,50]. Furthermore, the discrepancy could be exacerbated by permeability, efflux and free drug vs total drug availability, target location and biochemical state.

In summary, our results show that intracellular antibody-based selection of small compound ligands for specific proteins is practical and should provide a means to make compounds for disrupting abnormal protein function in any disease. Antibody fragments can be sequentially used for target validation and PPI epitope determination prior to drug development. Our approach allowed us to develop a series of compounds that bind to KRAS with high affinity, interfere with RAS PPI and inhibit RAS-dependent signalling in human tumour cells. This realizes the potential of allowing the replacement of antibody fragments (macrodrugs) with small molecule (conventional) potential drugs.

## Methods
**Fragment library**. A fragment library comprising 656 compounds, triaged for potential drug-like precursors, was used with molecular weight range from 94 to 341 Da, composed of chemical fragments that followed the "rule of three"

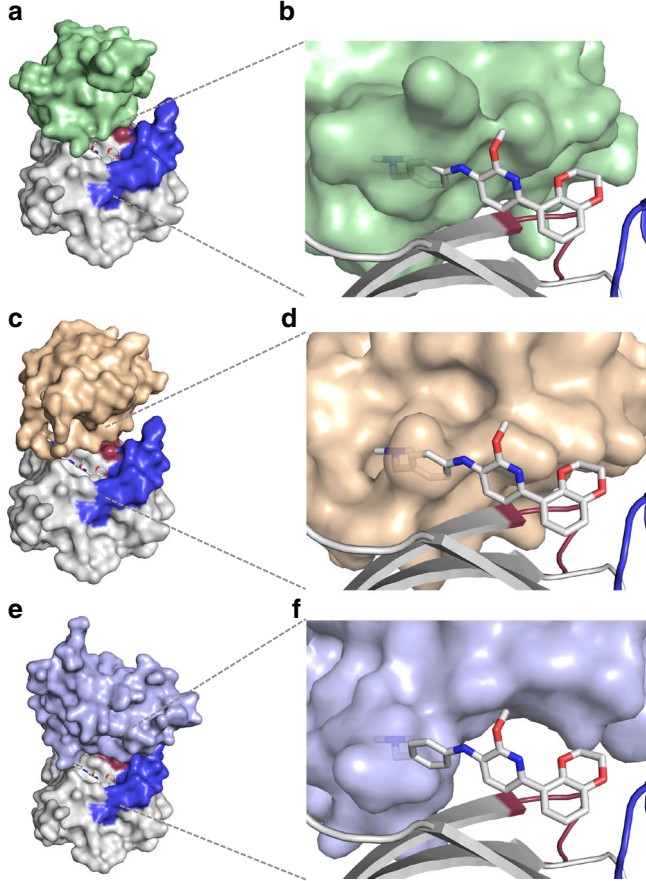

**Fig. 6** Superimposed structures of KRAS$_{169}$$^{Q61H}$-Abd-7 and RAS-effectors (RAF, Pi3K and RALGDS). The potential interactions that could prevent Abd-7 and a RAS effector binding simultaneously to the same KRAS molecule have been modelled by overlaying the structure of the KRAS$_{169}$$^{Q61H}$-Abd-7 complex onto published structures. In each case, simultaneous binding of KRAS to Abd-7 and the effector is sterically forbidden. **a, b** Overlay with HRAS-CRAF RBD (PDB 4G3X). Abd-7 would overlap with residues 62-67 of CRAF. **c, d** Overlay with HRAS-RALGDS RBD (PDB 1LFD). Abd-7 would overlap with residues 29-31 of RALGDS. **e, f** Overlay with HRAS-PI3Kγ RBD (PDB 1HE8). Abd-7 would overlap with residues 227-229 for PI3K

(molecular weight ≤ 300 Da, clogP ≤ 3, the number of hydrogen bond donors and acceptors each <3 and the number of rotatable bonds <3) and small molecules above 300 Da. Each compound was dissolved at 200 mM in 100% DMSO and 41 compounds were removed from the library because of insolubility in DMSO i.e. from the library, 615 useful compounds could be screened at the concentrations used. The library was replica plated (20 mM concentration) and stored at −80 °C. The initial hit compound (Maybridge) was *N*-(2-quinolinyl)-2-furamide (C$_{14}$H$_{10}$N$_2$O$_2$), product code HTS05481, MW:238.2456, clogP 2.33, Rotatable bonds: 2, H-bonds donor: 1, H-bonds acceptor 4.

**Recombinant protein expression for SPR and NMR**. Recombinant GST fusion RAS proteins were prepared by expression of HRAS (1-166) HRAS$^{G12V}$ or KRAS$^{G12V}$ cDNA cloned into the pGEX-2T vector in-frame with an N-terminal glutathione-s-transferase. pGEX-HRAS(wt) and pGEX-HRAS$^{G12V}$ plasmids were transformed into *E. coli* C41(DE3). Bacterial cells were cultured at 37 °C to an OD$_{600}$ of 0.6 and induced with IPTG (isopropyl 1-thio-beta-D-galactopyranoside, final concentration 0.1 mM) at 30 °C for 5 h. The bacteria cultures were harvested by centrifugation and the cell pellets re-suspended in 140 mM NaCl, 2.7 mM KCl, 10 mM NaH$_2$PO$_4$, 1.8 mM KH$_2$PO$_4$, 1 mM EDTA, 2 mM MgCl$_2$ pH 7.4. The proteins were extracted by cell disruption (Constant Systems Ltd., UK) at 25,000 psi at 4 °C. The GST-fusion proteins were purified by glutathione-sepharose column chromatography (GE Healthcare) and eluted with 50 mM Tris-HCl pH 8.0, 10 mM reduced glutathione, 1 mM DTT, 2 mM MgCl$_2$. The eluted proteins were dialysed against 50 mM Tris-HCl pH 8.0, 1 mM DTT, 2 mM MgCl$_2$ and concentrated to 10 mg/ml using a Biomax-30 ULTRAFREE-15 centrifugal filter device

(Millipore). To exchange endogenous guanidine nucleotide bound to RAS to GDP or GTP analogue, purified GST-RAS proteins were loaded with GTPγS, GppNHp or GDP (Sigma). Nucleotide exchange for SPR and NMR samples was done by preparing a final solution of 1 ml containing 400 ul of KRAS protein (final 0.63 mM), 94 µl of GPPNHP (16 times excess), 100 µl of alkaline phosphatase buffer [0.5 mM Tris Ph 8.5, 10 mM EDTA, 2 M (NH$_4$)$_2$SO$_4$], 100 µl of alkaline phosphatase (30 units/ml) and 307 µl of H$_2$O[51]. KRAS protein samples were concentrated using Vivapore 10/20 ml concentrator (7.5 kDa molecular weight cut-off; Sartorius Vivapore). The proteins were further purified by gel filtration on a HiLoad Superdex 75 10/300 GL column (GE Healthcare, Uppsala, Sweden) in a buffer containing 10 mM PBS pH 7.4, 5 mM MgCl$_2$ at a flow rate of 0.5 ml/min. Fractions corresponding to the protein were pooled and concentrated. Protein concentration was determined by extinction coefficient at 280 nm by using Prot-Param tool. Protein purity was analysed by SDS-PAGE stained with Instant Blue (Expedeon). For purification of anti-RAS scFv or VH, the plasmids pRK-HISTEV-scFv or pRK-HISTEV-VH were prepared by sub-cloning scFv or VH fragment into the pRK-HISTEV vector giving in-frame fusion with a 6x histidine tag and a TEV protease site. The plasmids were transformed into C41 (DE3), cultured at 37 °C to an OD$_{600}$ of 0.6 and induced with IPTG (final 0.5 mM) at 16 °C for 12 hours. scFv protein was extracted from bacteria pellets using sonication and a French press with the extraction buffer (25 mM Na phosphate, pH 7.4, 500 mM NaCl, 20 mM imidazole) and purified using His-Trap Ni-affinity columns (GE Healthcare) employing gradient elution (20–300 mM imidazole). The his-tag peptide was removed from the purified protein by treatment with TEV protease and dialysed in 20 mM Tris-HCl pH8.0, 300 mM NaCl, 20 mM imidazole at 4 °C overnight. The scFv was purified again by passing through a Ni-NTA agarose column (Qiagen) and by gel filtration on a HiLoad Superdex-75 HR column (GE Healthcare) in 20 mM Tris-HCl pH8.0, 250 mM NaCl and concentrated to 10 mg/ml for storage.

**Recombinant protein expression for crystallography**. KRAS$_{169}$$^{Q61H}$ and KRAS$_{188}$$^{G12D}$ cDNAs were cloned into the pRK-172 vector using NdeI and BamHI restriction sites, in-frame with an N-terminal 6x his-tag and the TEV protease recognition site. Plasmids containing pRK-172-KRAS$_{169}$$^{Q61H}$ and KRAS$_{188}$$^{G12D}$ sequences were transformed individually into B834(DE3)pLysS cells, which were grown in 25 ml LB medium with 50 µg/ml carbenicillin and 34 µg/ml chloramphenicol for 16 hours at 37 °C, before adding to 1 l LB medium containing the same antibiotics. Protein expression was induced when cells reached an OD$_{600}$ of 0.6 by addition of IPTG to a final concentration of 0.5 mM, followed by overnight incubation at 16 °C. Bacteria were harvested by centrifugation (5180 g, 30 min, 4 °C), resuspended in 60 ml of lysis buffer 50 mM Tris-HCl, pH 7.5, 500 mM NaCl, 5 mM MgCl$_2$ and 10 mM imidazole also containing one EDTA-free protease inhibitor cocktail table (Roche Diagnostics, Mannheim, Germany). Cells were lysed by sonication using five 30 s pulses at level of 16 and insoluble debris removed by centrifugation (75,600 g, 20 min, and 4 °C). The supernatant was applied to a column of nickel agarose beads (Invitrogen) by gravity, the beads were washed twice with lysis buffer containing 50 mM imidazole and bound proteins were eluted in 50 mM Tris-HCl, pH 7.5, 500 mM NaCl, 5 mM MgCl$_2$, 300 mM imidazole.

HIS-tagged TEV protease (1.4 mg/ml) was added to the KRAS$_{188}$$^{G12D}$ at a ratio of 1:100 to the eluate and the sample dialysed against 50 mM Tris-HCl, pH 7.5, 500 mM NaCl, 5 mM MgCl$_2$, overnight at 4 °C. The TEV protease and cleaved HIS-tag fragment were removed by re-application of the digestion to nickel agarose beads (Invitrogen). Samples containing KRAS protein were concentrated using Vivapore 10/20 ml concentrator (7.5 kDa molecular weight cut-off; Sartorius Vivapore). Endogenous guanidine nucleotide bound to RAS was exchanged to a GTP analogue and purified KRAS proteins were loaded with GppNHp (Sigma) following the same protocol described for the SPR and NMR experiments. The proteins were further purified by gel filtration using a Superdex 75 Increase 10/300 GL column (GE Healthcare, Uppsala, Sweden). KRAS$_{169}$$^{Q61H}$ was equilibrated with 20 mM HEPES pH 8.0, 150 mM NaCl, 5 mM MgCl$_2$ and 2 mM DTT and applied to the column at a flow rate of 0.5 ml/min. For KRAS$_{188}$$^{G12D}$ buffer used to equilibrate the column and elute the protein was 25 mM TrisCl, pH 8.0, 100 mM NaCl, 5 mM MgCl$_2$ and 1 mM TCEP. Fractions corresponding to KRAS were pooled and concentrated to 45–75 mg/ml for crystallisation trials. Protein concentration was determined from the extinction coefficient at 280 nm using ProtParam tool. Protein purity was confirmed by SDS-PAGE stained with Instant Blue (Expedeon).

**HRAS small-molecule screening and competitive SPR**. A BIAcore T100 (GE Healthcare) was used to screen the fragment library for RAS binders. A polyclonal goat anti-GST antibody (GE Healthcare) was immobilized on a CM5 sensor chip (GE Healthcare) by the amine coupling method. To immobilize the antibody on CM5 chip, the chip was first activated by injecting 100 µl mixture of 0.2 M EDC (N-ethyl-N-(dimethylaminopropyl) carbodiimide hydrochloride) and 0.05 M NHS (N-hydroxysuccinimide) at 10 µl/min flow rate. Furthermore, 100 µg/ml anti-GST antibody in 10 mM sodium acetate, pH 5.0 was injected at 5 µl/min for 900 s and immobilized until 20,000 ~ 25,000 RU. After immobilization the chip was immediately inactivated by injecting 1 M ethanolamine, pH 8.5 at 10 µl/min for 600 s. The chip was kept running in the HBS buffer (GE Healthcare) comprising 10 mM HEPES, pH 8.0, 150 mM NaCl. Of all, 5 µg/ml recombinant GST, GST-

HRAS$^{G12V}$-GTPγS or GST-HRAS$^{G12V}$-GDP proteins were injected into flow cell Fc4, Fc2 and Fc3, respectively for trapping through the immobilized anti-GST antibody. The GST-HRAS$^{G12V}$-GTPγS, GST-KRAS$^{G12V}$-GppNHpp, or GST-HRAS$^{G12V}$-GDP proteins were injected at 5 μl/min for 900 s and usually reached approximately 2000–2500 RU. The GST was injected at same flow rate until about 1200–1350 RU which is 0.54 times less than the GST-HRAS$^{G12V}$-GTPγS or GST-HRAS-GDP capture amounts, taking into consideration relative molecular weights (GST, 26KDa, GST-RAS, 48KDa, respectively). Before fragment library screening, the running buffer was replaced by 20 mM Tris, pH 8, 150 mM NaCl, 5 mM MgCl$_2$, 5% DMSO. DMSO solvent correction curves were generated by injecting the running buffers with serial concentrations of DMSO ranging from 4.5 to 6%. For fragment screening, each compound was diluted at 200 μM or 1 mM, 5% DMSO in 20 mM Tris, pH 8, 150 mM NaCl, 5 mM MgCl$_2$ buffer in a poly-propylene 96 well plate. The individual compounds were injected at 30 μl for 60 s at 30 μl/min flow rate over flow cells Fc1, Fc2, Fc3 and Fc4. The captured GST and GST-RAS were removed from the chip by rinsing with 40 μl 10 mM glycine-HCl, pH 2.1 at 20 μl/min flow rate after every 48 compounds sequential injection. The chip was regenerated by injecting new GST and GST-RAS proteins as described above.

For competitive SPR (cSPR) assays between anti-RAS scFv and candidate fragment binder, after immobilizing anti-GST polyclonal antibody on CM5 sensor chip as described above, then 5 μg/ml GST-HRAS-GTPγS protein was injected at 5 μl/min for 900 s to flow cells Fc2 and 4 and 5 μg/ml GST-HRAS-GDP to Fc3. Following GST-HRAS capture on the flow cells, 5 μg/ml anti-RAS scFv Y6-204 was injected at 5 μl/min for 900 s only to Fc4 until reaching an additional 1500–1800 RU, which is theoretically when all GST-HRAS-GTPγS should be bound to anti-RAS scFv with 1:1 interaction stoichiometry. After completing anti-RAS scFv injection, the sensograms were confirmed to be stable (i.e. anti-RAS scFv was not dissociated from GST-HRAS-GTPγS on Fc4). The candidate compounds were diluted at 15 μM to 2 mM serial concentrations in running buffer with final 5% DMSO, and a volume of 30 μl was injected for 60 s at a rate of 30 μl/min over all four flow cells of the CM5 chip.

The corresponding kinetic rate constants, $K_{on}$ and $K_{off}$, were determined using steady-state analysis of the compounds' binding affinities, assuming 1:1 ligand-protein stoichiometry. The BIAcore evaluation 2.1 software provided by the manufacture was used to analyse the data.

Calculations for the number of response units are required for a 1:1 ratio of compound/protein interaction are shown below.

Protein immobilization: 2500 RU; average fragment 250 approx. Da in size.
$R_{max}$ = (MWA/MWL) x RL x SM
MWA is the molecular weight of the analyte in Da
MWL is the molecular weight of the ligand in Da
RL is the immobilization level in RU
SM is the molar stoichiometry (assume 1:1)
$R$max = 250/47,500 × 2500 × 1
$R$max = 13 RU

## WaterLOGSY NMR

The waterLOGSY NMR method[35] was used to measure RAS ligand interaction[52]. WaterLOGSY experiments were conducted at a $^1$H frequency of 600 MHz using a Bruker Avance spectrometer equipped with a BBI probe. All experiments were conducted at 298 K, 3 mm diameter NMR tubes with a sample final volume of 200 μl was used in all experiments. Solutions were buffered using an H$_2$O 10 mM PBS, 5 mM MgCl$_2$ buffer corrected to pH 7.4. The sample preparation is exemplified as follows, the compound (10 μl of a 10 nM solution in DMSO-$d_6$) was added to an eppendorf before sequential addition of the H$_2$O PBS buffer (163.6 μl), D$_2$O (20 μl) and protein (6.4 μl, 311.8 μM). The resulting solution was spun to ensure full mixing and transferred to a 3 mm NMR tube before running the experiment. For a competition experiment using Y6-scFv or VH, the preparation was carried out in a similar manner; the compound (10 μl of a 10 mM solution in DMSO-$d_6$) was added to an eppendorf before sequential addition of the H$_2$O PBS buffer (146.4 μl), D$_2$O (20 μl), protein (6.4 μl, 311.8 μM) and Y6-ScFv (17.2 μl, 116.6 μM) with a ratio of 5.5:1:1. The resulting solution was spun and transferred to a 3 mm NMR tube before the run. Negative controls (compound alone, without the protein) were prepared in a similar manner, in order to obtain an end volume of 200 μl.

## $^1$H CPMG NMR

Typical experimental parameters for Carr–Purcell-Meiboom-Gill (CPMG) NMR spectroscopy were the following: total echo time, 40 ms; relaxation delay, 2 s; and number of transients, 264[53]. The PROJECT-CPMG sequence (90°x-[T-180°y-T- 90°y-T-180°y-T]$_n$-acq) was applied. Water suppression was achieved by pre-saturation. Prior to Fourier transformation, the data were multiplied with an exponential function with 3-Hz line broadening. The CPMG experiments were conducted at a $^1$H frequency of 700 MHz using a Bruker Avance with 5 mm inverse TCI 1 H/13 C/15 N cryo-probe. All experiments were conducted at 298 K and lapsed 128 scans. Furthermore, 3 mm diameter NMR tubes with a sample volume of 200 μl were used in all experiments. Solutions were buffered using a D$_2$O PBS buffer corrected to pH 7.4. The sample preparation is exemplified as follow: with 5 μM of protein (PBS pH 7.4, 5 mM MgCl$_2$), the compound (1.1 μl of a 10 mM (55 uM) solution in DMSO-$d_6$) was added to an eppendorf before sequential addition of the D$_2$O 10 mM PBS, 5 mM MgCl$_2$ buffer adjusted to pH 7.4 (194.0 μl)

and protein (4.9 μl, 205 μM, the protein is in an H$_2$O-based buffer for stability reason). The resulting solution was vortexed to be fully mixed and transferred to a 3 mm NMR tube before the run. Negative controls (compound alone, without the protein) were prepared in a similar manner, in order to obtain an end volume of 200 μl. CPMG protein titration experiments were carried out at a fixed compound concentration and a variable protein concentration. It was estimated that 55 μM solution of the compound was optimum for a good CPMG; protein was titrate increasingly from 0 μM until the signal of the compound was undetected in the proton NMR. The integrations of the protons acquired were compared to the reference (compound alone, 0 μM of protein) in order to obtain a percentage of decrease for each concentration of protein. Three different proton signals were used and a mean was calculated for each run. Each protein concentration was run in triplicate and a mean was calculated for each concentration. Where error bars are presented, they correspond to mean values ± SD of experimental repeats. Concentration and percentage of decrease were plotted and Kd fitting was run on the generated curve using Origin® with the following function: A*(1/(2*C))*((B + x + C)-sqrt(((B + x + C)^2)-(4*x*C))) where A is the maximum % of inhibition (i.e. 100), B is the Kd and C is the concentration of the compound which should not be fixed in the Origin® Equation due to potential variations in compound solubility.

## General experimental notes and instrument settings

All solvents and reagents were used as supplied (analytical or HPLC grade) without prior purification. Water was purified by an Elix® UV-10 system. Brine refers to a sat. aq. solution of sodium chloride. In vacuo refers to the use of a rotary evaporator attached to a diaphragm pump. Pet. ether refers to the fraction of petroleum spirit boiling between 30 and 40 °C, unless otherwise stated. Thin layer chromatography was performed on aluminium plates coated with 60 F254 silica. Plates were visualized using UV light (254 nm) or 1% aq. KMnO$_4$. Flash column chromatography was performed on Kieselgel 60 M silica in a glass column. NMR spectra were recorded on Bruker Avance spectrometers (AVII400, AVIII 400, AVIIIHD 600 or AVIII 700) in the deuterated solvent stated. The field was locked by external referencing to the relevant deuteron resonance. Chemical shifts (δ) are reported in parts per million (ppm) referenced to the solvent peak. $^1$H spectra reported to two decimal places, and $^{13}$C spectra reported to one decimal place, and coupling constants (J) are quoted in Hz (reported to one decimal place). The multiplicity of each signal is indicated by: s (singlet); br. s (broad singlet); d (doublet); t (triplet); q (quartet); dd (doublet of doublets); td (triplet of doublets); qt (quartet of triplets); or m (multiplet). Low-resolution mass spectra were recorded on an Agilent 6120 spectrometer from solutions of MeOH. Accurate mass measurements were run on either a Bruker MicroTOF internally calibrated with polyalanine, or a Micromass GCT instrument fitted with a Scientific Glass Instruments BPX5 column (15 m × 0.25 mm) using amyl acetate as a lock mass, by the mass spectrometry department of the Chemistry Research Laboratory, University of Oxford, UK.; m/z values are reported in Daltons.

## Analytical methods

Analysis of products and intermediates was carried out using reverse phase analytical HPLC-MS as well as normal phase analytical LCMS-MS. HPLC analytical methods: AnalpH2_MeOH_4 min: Phenomenex Luna C18 (2) 3 μm, 50 × 4.6 mm; A = water + 0.1% formic acid; B = MeOH + 0.1% formic acid; 45 °C; %B: 0.0 min 5%, 1.0 min 37.5%, 3.0 min 95%, 3.5 min 95%, 3.51 min 5%, 4.0 min 5%; 2.25 ml/min. AnalpH2_MeOH_QC_V1: Phenomenex Gemini NX C18 5 μm, 150 × 4.6 mm; A = water + 0.1% formic acid; B = MeOH + 0.1% formic acid; 40°C; %B: 0.00 min 5%, 0.5 min, 5%, 7.5 min 95%, 10.00 min 95%, 10.1 min 5%, 13.00 min 5%; 1.5 ml/min. LCMS Analytical Methods: LCMS Agilent_Gen Method A [100–1000] 6 MIN: Phenomenex Gemini NX C18 5 μm, 150 × 4.6 mm; A = water + pH 9 (Ammonium Bicarbonate 10 mM); B = MeOH; 40°C; %B: 0.0 min 5%, 0.50 min 5%, 7.5 min 95%, 10.0 min 95%, 10.1 min 5%, 13.0 min 5%; 1.5 ml/min.

$^1$H and $^{13}$C NMR spectra of Abd-2 to Abd-7 and LCMS data for Abd-2, Abd-2a, Abd-2b are shown in Supplementary Figs. 9–16. Chemical synthesis protocols are available in Supplementary notes 1–7.

## X-ray crystallography

For X-ray diffraction experiments, KRAS$^{Q61H}$-GppNHp crystals were grown at 4 °C from 1.5 ± 1.5 μl vapour diffusion drops containing 75 mg/ml KRAS$^{Q61H}$, 8–15% w/v Polyethylene Glycol 3350 and 0.2 M lithium citrate pH 5.5 giving crystal form I. For KRAS$^{G12D}$–GppNHp crystals, drops were prepared by mixing 1.5 μl of protein at 45 mg/ml with 1.5 μl of reservoir consisting of 0.1 M TrisCl pH 8.0, 0.2 M NaOAc and 32% PEG 4000 in 24-well Cryschem sitting-drop plates giving crystal form II. For data collection, crystals were cryo-protected by addition of 20% glycerol and flash-cooled in liquid nitrogen. For crystal soaking experiments, compounds were added individually (25–50 mM or saturated solution of compound with a final DMSO concentration of 6–12% v/v) to the crystallisation buffer. Crystals were then transferred to solution containing compound for a minimum of 5 min. Soaked crystals were cryo-protected with 20% glycerol and flash-cooled in liquid nitrogen for data collection. In each case, X-ray diffraction data were collected at the Diamond Light Source (DLS, Oxfordshire). The structure of KRAS$^{Q61H}$ GppNHp–Abd2 to Abd-7 and KRAS$^{G12D}$ GppNHp-compound Abd-3 were solved by molecular replacement using Protein Data Bank

(PDB) codes 3GFT and 4DSU, respectively, with the programme Phaser[54,55]. Structures were refined using REFMAC in the CCP4 suite[56] and manually corrected using Coot[57]. Crystal form I (KRAS[Q61H]) has six KRAS molecules in the asymmetric unit, assembled as a hexamer. Electron density maps averaged with six-fold non-crystallographic symmetry (NCS) were used to improve the definition of the bound compounds. Local ncs averaging within REFMAC was used throughout refinement and the final models were validated using PROCHECK molPROBIT and Phenix software packages[58,59]. Figures were created using PyMOL[60]. Data collection and refinement statistics are summarized in Supplementary Tables 1 and 2.

The PBD codes for the structures presented in this paper are:

KRAS$_{169}^{Q61H}$ GPPNHP-Abd-2 PDB ID: 5OCO
KRAS$_{169}^{Q61H}$ GPPNHP-Abd-3 PDB ID: 5OCT
KRAS$_{188}^{G12D}$ GPPNHP-Abd-3 PDB ID: 5OCG
KRAS$_{169}^{Q61H}$ GPPNHP-Abd-4 PDB ID: 6FA1
KRAS$_{169}^{Q61H}$ GPPNHP-Abd-5 PDB ID: 6FA2
KRAS$_{169}^{Q61H}$ GPPNHP-Abd-6 PDB ID: 6FA3
KRAS$_{169}^{Q61H}$ GPPNHP-Abd-7 PDB ID: 6FA4

**Kinase panel binding assay.** Experiments were performed by Eurofins Pharma Discovery Services. The KinaseProfiler™ assay protocol guide can be downloaded from their website: www.eurofins.com/discoveryservices. The guide contains information on the buffer conditions, substrate and reference inhibitor blank used for each kinase.

**Tissue culture assays.** HEK293T human embryonic kidney cells were grown in DMEM medium (Life Technologies) supplemented with 10% FBS (Sigma) and 1% Penicillin/Streptomycin (Life Technologies). Cells were grown at 37 °C with 5% CO$_2$. Cells were obtained for the American Tissue Culture Collection except HEK293T that was obtained locally.

**BRET2 cell assay.** The BRET assay was carried out as described elsewhere[42]. A total of 650,000 HEK293T were seeded in each well of a six well plate. Approximately, 24 h later, cells were transfected with an appropriated BRET-based RAS biosensor (i.e. RAS-effector) using Lipofectamine 2000 transfection reagent (Thermo-Fisher). Cells were detached 24 h later and washed with PBS and seeded in a white 96 well plate (clear bottom, PerkinElmer, cat#6005181) in OptiMEM no phenol red medium (Life Technologies) complemented with 4% FBS. Cells were left for 4 h at 37 °C before adding compounds. Stock compounds were held at 10 mM in 100% DMSO and diluted in OptiMEM no red phenol + 4% FBS to reach 10X the final concentration (2% DMSO for each concentration). The final concentrations in the cells were 0, 5, 10 and 20 µM (therefore the intermediate 10X concentrations were 0, 50, 100 and 200 µM. Furthermore, 10 µl of 10X compounds were added in each well of the 96 well plate to reach 0, 5, 10 and 20 µM final concentrations (with final 0.2% DMSO each). Quadruplicates were performed for each point. Cells were left for an additional 20 h at 37 °C before the BRET2 signal reading directly after addition of Coelenterazine 400a substrate (10 µM final) to cells (Cayman Chemicals, cat#16157). BRET2 reading was carried out on an Envision instrument (2103 Multilabel Reader, PerkinElmer) with the BRET2 Dual Emission optical module (515 nm ± 30 and 410 nm ± 80; PerkinElmer).

**Biomarker phosphorylation assay.** A total of 450,000 DLD1 or 640,000 H358 cells were seeded per well of a six well plate. Approximately, 20 h later, the medium was removed, the cells washed twice with PBS and DMEM without FBS was added and left for 24 h. The medium was removed and 2 ml of DMEM no FBS was added with the appropriate concentration of compound (DMSO, 2, 5, 10 and 20 µM) for 3 h. Cells were stimulated with EGF (50 ng/ml) for 10 min, washed once with PBS and directly lysed in SDS-Tris buffer (1% SDS, 10 mM Tris-HCl pH 7.4) supplemented with protease inhibitors (Sigma) and phosphatase inhibitors (Thermo-Fisher). Cell lysates were sonicated with a Branson Sonifier and the protein concentrations determined by using the Pierce BCA protein assay kit (Thermo-Fisher). Furthermore, 10 µg of protein was resolved on 10 or 12.5% SDS-PAGE and subsequently transferred onto a PVDF membrane (GE). The membrane was blocked (with 10% non-fat milk (Sigma) or 10% BSA (Sigma) in TBS-0.1% Tween20) and incubated overnight with primary antibody at 4 °C. Primary antibodies include anti-phospho-p44/22 MAPK (ERK1/2) (1/4000, CST, Cat#9101S), anti-p44/42 MAPK (total ERK1/2) (1/1000, CST, Cat#9102S), anti-phospho-AKT S473 (1/1000, CST, Cat#4058S), anti-AKT (1/1000, CST, Cat#9272S), anti-cyclophilin B (1/1000, Abcam, Cat#178397). After washing, the membrane was incubated with an anti-rabbit IgG HRP-linked (1/3000, CST, Cat#7074S) secondary antibody for 1 h at room temperature. The membrane was washed with TBS-0.1% Tween and developed using Pierce ECL Western Blotting Substrate (Thermo-Fisher) and CL-XPosure films (Thermo-Fisher). Uncropped Western blots are shown in Supplementary Fig. 17.

**Cell viability assays.** Cancer cell lines were in seeded in ViewPlates-96 microplates (PerkinElmer). DLD-1 cells were plated at 10,000 cells per well and HT1080 cells at 7500 cells per well and cultured in DMEM, high glucose, GlutaMAX media containing 10% FBS at 37 °C in 5% CO$_2$ atmosphere. Cells were cultured overnight and the compounds (dissolved in DMSO and diluted to 0.2% DMSO) were added

to the cells at concentrations ranging from 0 to 20 µM. The cells were incubated under standard culture conditions for either 24, 48 or 72 h. Cell viability was as quantitated using the CellTiterGlo Luminescent Cell Viability Assay (Promega) according to the manufacturer's instructions to measure ATP generated by metabolically active cells. Luminescent signals were measured using an Envision 2103 Multilabel Microplate Reader (PerkinElmer). The luminescence signals obtained from the compound-treated cells were normalized against the signal for DMSO-only treated cells. The IC50 values, calculated from the 48 and 72 h CellTiterGlo data, were generated by non-linear regression using the software GraphPad Prism 7.00 for windows (GraphPad Inc).

**Data availability.** Crystallographic data have been deposited in the PDB, accession codes: 5OCO, 5OCT, 5OCG, 6FA1, 6FA2, 6FA3, 6FA4. Other relevant data of this study are included within the article and its Supplementary Information. All data supporting the findings of this study are also available from the corresponding author upon reasonable request.

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

## Acknowledgements

This work was supported by grants from the Wellcome Trust (100842/Z/12/Z) and (099246/Z/12/Z), the Medical Research Council (MR/J000612/1) and Bloodwise (12051). We are grateful to Dr Martin Drysdale and the CRUK Beatson Institute Drug Discovery Group for providing the chemical fragment library and to Professor Andy Hamilton for discussions on the project. We would like to thank Prof. Tim D.W. Claridge for help and crucial advice on NMR techniques. We also like to thank Dr. Tim Fagge for very helpful advice on the use of the Biacore T100 and T200. We acknowledge the beamline staff at stations I02, I03 and I04-1 at the Diamond Light Source, UK (proposals mx9306 and mx13456) and the European Synchrotron Radiation Facility at stations ID30A-1 and ID30A-3 and we would like to thank Matthew Bowler and Edward Lowe for assistance during X-ray diffraction experiments. We also wish to thank Drs. Stuart Crosby, Jonathan Dunn, Robert Freem, Traore Tenin and Sophie Williams for the chemistry. We are also grateful to Drs. Trevor Perrior, Alan Naylor and Graham Showell for input and advice.

## Author contributions

Originator of project: T.R. Conducted experiments: C.Q., A.C., T.T., D.P., N.B., L.L., A.M., C.B., H.T., M.E. Contributed new reagents or analytic tools: N.F.F. Performed data analysis: C.Q., A.C., N.B., T.T., D.P., C.B., L.L., P.F., M.E., N.F.F., A.R., S.C., S.P., T.R. Contributed to the writing of the manuscript: All authors.
