## [Peer Review File · Nature Communications]

Reviewers' Comments:

Reviewer #1:

Remarks to the Author:

The paper under review can be considered another example of compound identification for interference with Ras-effector interactions by means of fragment-based and subsequent structure guided compound design. This approach has been published before by the Genentech and Novartis teams as well as several academic groups. The difference here is the use of an antibody technology to identify the first hit series. The identified binding site is the same as targeted in the approaches mentioned above.

The paper then also suffers from the same problems as the earlier findings. There is a major disconnect between the *in vitro* activities and the *in cellulo* activities. In the case under review here, the difference is ca. 100fold (!). In addition, influence on signalling pathways is weak and only clear at higher concentrations (e.g. phosphorylation in the Mek/Erk pathway). The developments in industry ultimately were given up due to these problems.

Taken together, the paper under review does report a new method to arrive at initial fragments, but it does not go beyond the state of the art in the other points raised. For acceptance in a high profile journal one would want to see novelty, i.e. the discovery of a new binding site, or clear cellular activity at low concentration, a clear difference between Ras-mutated and non-mutated cells, and ultimately clear activity in mouse xenografts.

This paper is scientifically certainly interesting and should be published in a good med chem journal. Acceptance for publication in Nature Communications is not indicated.

Reviewer #2:

Remarks to the Author:

This paper describes the development of chemical drug leads against activated mutant RAS, for treatment of cancer. It provides spectacular proof of concept for what the authors argue is a general method of drug development against intracellular targets in which the biology is mediated by protein-protein interactions. Although none of the elements in the method are truly novel, they have been combined here in a novel and effective way.

(a) The authors started with a single domain antibody (VH), which they had derived earlier against mutant activated RAS and had shown to block RAS interactions with effector proteins in cells.

(b) The authors then screened a small library of small chemical fragments for their ability to compete with the VH/RAS complex, identifying one compound that bound to activated mutant but not controls.

(c) The authors then created a series of chemical leads with improved properties derived from the first compound. The creation of the new leads was guided using the X-ray crystallographic structures of each lead in complex with mutant activated RAS, and screened as in (b). This allowed the generation of a lead with binding affinity of 38 nM, that was both soluble and cell permeable. Indeed the lead was able to block RAS protein-protein interactions in cell lines, and to compromise the viability of the cells.

As stated earlier none of the individual elements are truly novel in isolation. In (a) the intracellular VH against mutant activated RAS had already been described by the authors. Furthermore several single domain antibodies have already been made against a variety of proteins, including receptor-ligands pairs, and shown to be capable of blocking such interactions. In (b) the use of screening by competition assays, including the use of antibodies or a protein or macrocyclic ligand to "mark" a target site is routine in the pharmaceutical industry, as is (c) in which X-ray crystallography and structure-aided design is used to elaborate leads.

However the assembly of these elements (a) + (b) + (c) does seem to be novel for RAS, indeed for any intracellular protein target. The main novelty comes with (a) + (b) - the expression of antibody domains known to block the target biology within cells, and their use for drug screening. Not only does this allow identification of those regions of the target that mediate biology within the cell, it also helps narrow the focus to the small portion of the target surface that is relevant. [In principle, instead of antibody single domains, it should also be possible to make other small intracellular domains from phage display libraries for the same purpose].

It is worth making comparison with other approaches - for example macrocycle libraries (Peptidreams) can be used to generate reagents in vitro to "pockets" on a target, and used in competition assays to identify compounds binding to the same pockets. However this begs the question of whether such pockets are relevant to the target biology. Here this matter is established at the outset. Another method for drug screening might involve the screening of huge libraries of chemical compounds directly on cells, with a cell-based readout. In the method described here, only a small library was used, and the first and only "hit" could only be detected using a competition assay, but was not detected using a cell-based screen.

Although the novelty may seem to be subtle, the proof of concept described here validates the approach, and seems to have a practical edge over several other methods for generation of drug leads against intracellular targets. I therefore recommend that it should be accepted for publication.

However I do have some issues with the paper. I think it could have been written in a less technical manner, and I would have preferred less space used on the rationale for building the new chemical leads, and more space on identifying the novelty and potential utility compared with other methods, as indeed I have tried to do above.

A couple of queries:

Line 72 and 73.

Not sure what this means ? Surely this doesn't limit the region on the target - in a competition assay the small compound could also bind partly outside this region ? Indeed see Fig.2 where Abd-2 extends a fraction beyond the region of the target covered by VH.

Line 102.

The chemical compound library, which comprises only 656 compounds seems very small to have been so successful - was this library pre-selected in any other way than suggested in the methods lines 365-371 ? Can the authors describe the diversity of molecular species in this chemical library ? Do the authors think that their method of detection (with its threshold in the 100 micromolar range) facilitates the use of very small chemical libraries ?

Fig 1a-c seem over-elaborate to describe a simple competition assay using SPR.

Reviewer #3:

Remarks to the Author:

In this manuscript, Quevedo et al describe a series of K-Ras inhibitors derived from small molecule Abd-1 that competes with an antibody scFv binding site on K-Ras. The approach of using SPR for competition of a small molecule that binds to the same epitope recognized by an antibody is interesting, however it yielded a binding site that has already been well studied. Small molecules that bind to the groove between switch I and II have been well characterized by the Fesick lab (Sun et al 2012), Genentech (Maurer et al 2012), and Astrazeneca (Winter et al 2015). These studies have in fact have derived inhibitors that make similar water-mediated interactions to D54 and S39. In considering the novelty of the cSPR approach and inhibitors, the inhibitor series is very similar to those studied in the past. However, what the authors do achieve is cellular activity for their compounds which only one other study has done to date (Jansen et al 2017). Through structure-activity-relationships that were extensively verified with X-ray crystallography the authors were able to improve the series to obtain Abd-7. They demonstrated that Abd-7 has cellular activity at micromolar concentrations in a K-RasG13D and a NRASQ61K cell lines, showing potential for inhibition across Ras isoforms and mutations.

Specific comments

1. It is not clear how was the anti-Ras scFV chosen. Brief information should be included describing the original study where the antibody was characterized, its binding site, and the crystallographic evidence.
2. The western blots appear to show that pAKT is inhibited more readily than pERK by Abd-7. Any theories for why that is so based on the model in Fig 6?
3. The potencies for Abd-5, -6, and -7, measured by NMR are 220nM, 38nM, and 51nM respectively. However the concentrations used in the KRAS/PI3K, and other effector binding interactions are 5, 10, and 20 microM. This is a significant mismatch, and the effects on disruption of PI3K shown in Figure 4A, show that even at 20microM, 50% of the signal has not been inhibited.
4. Another confusing aspect of the data presented is related to the cellular data in which 10-20 microM of Abd7 is sufficient in different cellular systems to block P-Akt completely, which is not consistent with the data in Figure 4A. The authors must address inconsistencies in the biochemical and cellular data relating to the Abd series of inhibitors.

Response to Reviewer 's Comments

Reviewer 1

The paper under review can be considered another example of compound identification for interference with Ras-effector interactions by means of fragment-based and subsequent structure guided compound design. This approach has been published before by the Genentech and Novartis teams as well as several academic groups. The difference here is the use of an antibody technology to identify the first hit series.

Author comment: A key novel element of our work is that intracellular antibodies can be used initially for target validation and subsequently to initiate small molecule drug development. This is novel in a general applicable context and novel in the context of KRAS. Regarding the novelty of the compounds, while there are examples in literature of RAS-effector inhibition but these are peptides (Upadyaya,) and lack of crystal structure, or show indirect effects on effector response via interference of RAS-SOS interaction (e.g. Authors Fesik, Winter) or non-RAS-effector interaction despite observing some RAS:SOS interaction modulation (Authors Genentech). Several other groups show a direct RAS-effector inhibition but lack of crystal structure-based design and rational. Our approach combines the hit identification via a novel method, structure-based design via crystal structures and NMR methodologies with proof of concept in mechanistic and cell assays.

We have amended the text to articulate these points (lines 309 to 323).

The identified binding site is the same as targeted in the approaches mentioned above

Author comment: Our previously published HRAS-scFv crystal structure, which pre-dates all the randomly selected small molecule work, binds in this region of RAS, so inevitably, the antibody derived compounds also bind in this region. Further, as reviewer 3 comments "Through structure-activity-relationships that were extensively verified with X-ray crystallography the authors were able to improve the series to obtain Abd-7"

The binding site of our compounds was initially identified by Gorfe et al and demonstrated by Genentech with DCAI and another two fragments but not explored for the development of compounds with direct inhibition of RAS-effector interactions. Other groups have demonstrated that their compounds bind to that region but lack of X-ray data to develop and strengthen SAR and potent compounds. With our X-ray crystallography, we were able to demonstrate that high affinity binders can be identified without the need of covalent binding (one of the problems of RAS modulation by small molecules) and that the known pocket can be further explored to design of RAS-effector inhibitors.

We have amended the text (lines 330 to 336).

The paper then also suffers from the same problems as the earlier findings. There is a major disconnect between the *in vitro* activities and the *in cellulo* activities. In the case under review here, the difference is ca. 100fold (!). In addition, influence on signalling pathways is weak and only clear at higher concentrations (e.g. phosphorylation in the Mek/Erk pathway).

Author comment: This is an important issue but drop-off in potency between *in vitro* and *in vivo* is common in drug development because *in vitro* binding assays are done in pure solutions of proteins rather than the complex mixture of proteins, membranes, nucleic acids, lipids etc that exist in cells. Further, RAS proteins are located on the inside of the plasma membrane to which the compounds must access for their action. However, we do find (Figure 5) that there is a direct correlation between SAR, *in vitro* K_d and *in vivo* IC₅₀ so that while there is drop-off in potency in cells compared to *in vitro*, there is increased cell potency as the binding changes. This is the first time that such an effect has been reported and shows that future generations of this chemical series can be modified to minimise the drop-off. A clear demonstration of this is Abd-6 and Abd-7, obtained via chemistry development, with similar K_d against KRAS but with different IC₅₀ in cell viability assays. Further, it has been reported in the literature that allosteric binding in very flexible protein targets have the same high drop off between *in vitro* and *in cellulo* (Lawson, ref. 49).

We have added a further part in the discussion about this issue (lines 378 to 383).

The developments in industry ultimately were given up due to these problems.

Author comment: Certainly, we plan to address these complicated issues with a next generation of compounds since we can explore the crystal structures that we developed for this series and which is something that could not be addressed so far in other published work and, presumably industry. We have included a comment (lines 378 to 383).

Taken together, the paper under review does report a new method to arrive at initial fragments, but it does not go beyond the state of the art in the other points raised. For acceptance in a high profile journal one would want to see novelty, i.e. the discovery of a new binding site, or clear cellular activity at low concentration, a clear difference between Ras-mutated and non-mutated cells, and ultimately clear activity in mouse xenografts.

Author comment: Rather we would argue that we have reported novelty in our new methods of cSPR screening, using high affinity antibody, for initial compounds from libraries (agreed by referee 2) and further we do indeed “go beyond the state of the art” because we have shown, SAR based on high resolution crystal structures of KRAS-GTP, with associated compounds not fragment hits (that has not been done in any of “the earlier findings”) and increased potency both *in vitro* binding (Kd) and in cell-based assays where IC₅₀ improve with a correlation to Kd. This has not been shown before and thus new state of the art.

This paper is scientifically certainly interesting and should be published in a good med chem journal. Acceptance for publication in Nature Communications is not indicated.

Reviewer 2

This paper describes the development of chemical drug leads against activated mutant RAS, for treatment of cancer. It provides spectacular proof of concept for what the authors argue is a general method of drug development against intracellular targets in which the biology is mediated by protein-protein interactions. Although none of the elements in the method are truly novel, they have been combined here in a novel and effective way.

Author comment: As above comment to referee1, we would argue that we have reported novelty in our new methods of cSPR screening, using high affinity antibody, for initial compounds from libraries and further we go well beyond the “state of the art” because we have shown high resolution crystal structures of KRAS-GTP with associated compounds (that has not been done in any of “the earlier findings”) and increased potency both *in vitro* binding (Kd) and in cell-based assays where IC₅₀ improve with a correlation to Kd. This has not been shown before and thus new.

We have added to the discussion the novelty of our approach (lines 309 to 323).

(a) The authors started with a single domain antibody (VH), which they had derived earlier against mutant activated RAS and had shown to block RAS interactions with effector proteins in cells.

(b) The authors then screened a small library of small chemical fragments for their ability to compete with the VH/RAS complex, identifying one compound that bound to activated mutant

(c) The authors then created a series of chemical leads with improved properties derived from the first compound. The creation of the new leads was guided using the X-ray crystallographic structures of each lead in complex with mutant activated RAS, and screened as in (b). This allowed the generation of a lead with binding affinity of 38 nM, that was both soluble and cell permeable. Indeed the lead was able to block RAS protein-protein interactions in cell lines, and to compromise the viability of the cells.

As stated earlier none of the individual elements are truly novel in isolation. In (a) the intracellular VH against mutant activated RAS had already been described by the authors.

Furthermore several single domain antibodies have already been made against a variety of proteins, including receptor-ligands pairs, and shown to be capable of blocking such

interactions. In (b) the use of screening by competition assays, including the use of antibodies or a protein or macrocyclic ligand to “mark” a target site is routine in the pharmaceutical industry, as is (c) in which X-ray crystallography and structure-aided design is used to elaborate leads.

However the assembly of these elements (a) + (b) + (c) does seem to be novel for RAS, indeed for any intracellular protein target. The main novelty comes with (a) + (b) - the expression of antibody domains known to block the target biology within cells, and their use for drug screening. Not only does this allow identification of those regions of the target that mediate biology within the cell, it also helps narrow the focus to the small portion of the target surface that is relevant. [In principle, instead of antibody single domains, it should also be possible to make other small intracellular domains from phage display libraries for the same purpose].

It is worth making comparison with other approaches - for example macrocycle libraries (Peptidreams) can be used to generate reagents in vitro to “pockets” on a target, and used in competition assays to identify compounds binding to the same pockets. However this begs the question of whether such pockets are relevant to the target biology. Here this matter is established at the outset. Another method for drug screening might involve the screening of huge libraries of chemical compounds directly on cells, with a cell-based readout. In the method described here, only a small library was used, and the first and only “hit” could only be detected using a competition assay, but was not detected using a cell-based screen.

Author comment: we would like to stress that our use of the intracellular antibody against mutant activated RAS was done because, at that time, it had not been formally proved that inhibiting RAS-effector PPI would affect tumour growth. Our work showed this was the case, thus allowing us to use the intracellular VH to develop compounds as drug leads.

Although the novelty may seem to be subtle, the proof of concept described here validates the approach, and seems to have a practical edge over several other methods for generation of drug leads against intracellular targets. I therefore recommend that it should be accepted for publication.

However I do have some issues with the paper. I think it could have been written in a less technical manner, and I would have preferred less space used on the rationale for building the new chemical leads, and more space on identifying the novelty and potential utility compared with other methods, as indeed I have tried to do above.

Author comment: In response, we have added to the discussion on the novelty of our approach (lines 309 to 323).

A couple of queries:

Line 72 and 73.

Not sure what this means? Surely this doesn't limit the region on the target - in a competition assay the small compound could also bind partly outside this region? Indeed see Fig.2 where Abd-2 extends a fraction beyond the region of the target covered by VH.

Author comment: we meant that a single domain (VH or VL) would contact a small region and would occupy less surface than a putative compound of less than 500 daltons; this means that we would not need very large compounds to occupy the same regions of the Dab. We have amended the text to make this more explicit. Please see lines 72 to 76

Line 102

The chemical compound library, which comprises only 656 compounds seems very small to have been so successful *Author comment: We obtained 26 initial hits, of which one was of interest (see new comment line 110)* - was this library pre-selected in any other way than suggested in the methods lines.

Author comment: the fragment library was only triaged for potential drug-like precursors. We have amended the methods to add this point⁴

Can the authors describe the diversity of molecular species in this chemical library ? Do the authors think that their method of detection (with its threshold in the 100 micromolar range) facilitates the use of very small chemical libraries ?

Author comment: Ideally, future use of this method would be better carried out with larger, diverse libraries as our library was small and initially we obtained 26 check RAS-binders but only one had the property that we wanted (i.e. did not bind when the VH contacts the RAS protein). This is amended and noted in lines 105 and 110

Fig 1a-c seems over-elaborate to describe a simple competition assay using SPR.

Author comment: While we agree, this figure is elaborate, the competition is not simply competition but also allowing us to distinguish from HRASG12V-GTP from HRASG12V-GDP and of course HRASG12V-GTP-scFv complex. We prefer to keep this figure.

Reviewer III

In this manuscript, Quevedo et al describe a series of K-Ras inhibitors derived from small molecule Abd-1 that competes with an antibody scFv binding site on K-Ras. The approach of using SPR for competition of a small molecule that binds to the same epitope recognized by an antibody is interesting, however it yielded a binding site that has already been well studied. Small molecules that bind to the groove between switch I and II have been well characterized by the Fesick lab (Sun et al 2012), Genentech (Maurer et al 2012), and Astrazeneca (Winter et al 2015). These studies have in fact have derived inhibitors that make similar water-mediated interactions to D54 and S39. In considering the novelty of the cSPR approach and inhibitors, the inhibitor series is very similar to those studied in the past. However, what the authors do achieve is cellular activity for their compounds which only one other study has done to date (Jansen et al 2017). Through structure-activity-relationships that were extensively verified with X-ray crystallography the authors were able to improve the series to obtain Abd-7. They demonstrated that Abd-7 has cellular activity at micromolar concentrations in a K-RasG13D and a NRASQ61K cell lines, showing potential for inhibition across Ras isoforms and mutations.

Specific comments

1. It is not clear how was the anti-Ras scFv chosen.

Brief information should be included describing the original study where the antibody was characterized, its binding site, and the crystallographic evidence.

Author comment: The anti-RAS intracellular antibody against mutant activated RAS was done because, at that time, it had not been formally proved that inhibiting RAS-effector PPI would affect tumour growth. Our work showed this was the case, thus allowing us to use the intracellular VH to develop compounds as drug leads. We have extended the introduction to make this more explicit (lines 88 to 90). We feel that the reasons for choosing the anti-Ras scFv were explained (lines 88 to 94).

2. The western blots appear to show that pAKT is inhibited more readily than pERK by Abd-7. Any theories for why that is so based on the model in Fig 6?

Author comment: This is an intriguing finding and we do not wish to hypothesise at present. However, it appears that the overlap of binding sites that could produce the steric clash are somewhat dissimilar. To make this potential clash clearer, we have produced a new version of Figure 6 in which the overlays are transparent showing more clearly the regions of overlap. However, we believe that despite the same level of steric disruption observed with our compounds, the competition with the RAS-effectors has a different outcome probably associated with the affinity of the different effectors, for instance PI3K has an estimated affinity of 3.2 μ M while RAF has an affinity of 0.08 μ M (Erijman)

We have added a further part in the discussion (lines 362 to 366).

3. The potencies for Abd-5, -6, and -7, measured by NMR are 220nM, 38nM, and 51nM respectively. However, the concentrations used in the KRAS/PI3K, and other effector binding interactions are 5, 10, and 20 microM. This is a significant mismatch, and the effects on disruption of PI3K shown in Figure 4A, show that even at 20microM, 50% of the signal has not been inhibited.

Author comment: As discussed for referee 1's comments, this is an important issue but drop-off in potency between *in vitro* and *in vivo* is common in drug development because compound *in vitro* binding assays are done in pure solutions of proteins rather than the complex mixture of proteins, membranes, nucleic acids, lipids etc that exist in cells. Further, the RAS protein is located on the inside of the plasma membrane to which the compounds must access for their action. However, we do find (Fig 5) that there is a direct correlation between SAR, *in vitro* Kd and *in vivo* IC₅₀ so that while there is drop-off in potency in cells, there is increased potency as the binding affinity improves. This is the first time that such an effect has been reported and shows that future generations of this chemical series can be modified to minimise the drop-off. Figure 4A shows BRET results of transient transfected assays where the level of disruption of the RAS-effector is in a qualitative and not quantitative manner since, in transient transfected assays, the quantities of protein expressed are difficult to control. Nonetheless, several of the BRET inhibition results at 20 uM are more than 50% inhibited.

A short description has been added (lines 254 to 257 and lines 378 to 383).

4. Another confusing aspect of the data presented is related to the cellular data in which 10-20 microM of Abd7 is sufficient in different cellular systems to block P-Akt completely, which is not consistent with the data in Figure 4A. The authors must address inconsistencies in the biochemical and cellular data relating to the Abd series of inhibitors.

Author comment: The assays are carried out with different parameters and it makes correlation hard to achieve in a meaningful way, e.g. the effect of compounds on endogenous pAKT and pERK is done at 2 hours whereas viability is after 5 days. Further the BRET assays are done with transient transfected cells (Fig 4) in which concentrations are hard to normalise. We have added a part in the text to emphasise this point and, we hope, add clarity to the comparison between these assays.

As above, a short description has been added (lines 254 to 257).